# The Effect of Three Surgical Therapies to Increase Keratinized Mucosa Surrounding Dental Implants with Peri-Implantitis: A Pilot Study

**DOI:** 10.3390/medicina57101093

**Published:** 2021-10-12

**Authors:** In-Kyung Lee, Hyun-Seok Choi, Sang-Heon Jeong, Jung-Tae Lee

**Affiliations:** 1Department of Periodontics, Jukjeon Dental Hospital, College of Dentistry, Dankook University, Yongin 16890, Korea; perio8296@dankook.ac.kr (I.-K.L.); futurechoi0621@gmail.com (H.-S.C.); 2Department of Radiology, Jukjeon Dental Hospital, College of Dentistry, Dankook University, Yongin 16890, Korea; serch34@hanmail.net; 3Department of Periodontics, One-Stop Specialty Center, Seoul National University, Dental Hospital, Seoul 05698, Korea

**Keywords:** dental implantation, peri-implantitis, hard palate, oral mucosa, tissue grafts

## Abstract

*Background and Objectives:* The purpose of this pilot study was to evaluate the clinical outcomes of three different methods for increasing the keratinized mucosa (KM) surrounding dental implants with peri-implantitis. *Materials and methods:* Twenty implant sites with peri-implantitis were divided into: (1) porcine collagen matrix (CM) group: seven implant sites; (2) apically positioned flap (APF) group: eight implant sites; and (3) free gingival graft (FGG) group: five implant sites. The KM width and clinical parameters (probing pocket depth (PPD) and bleeding on probing (BOP)) were measured at time points: before surgery (T0) and 30 (T1), 60 (T2), 90 (T3), and 180 (T4) days after surgery. *Results:* Regarding KM width, all the groups had significant differences for increasing horizontal and vertical KM width. The CM and FGG groups had greater KM than the APF group. There was a decrease in PPD in all three groups. APF and FGG showed significant differences in PPD at T1 and T2 compared to T0. Only the FGG group showed a significant difference in PPD at T3 and T4 compared with that at T0. BOP values were also reduced in all the groups at T1–T4 compared to T0. The APF and FGG groups showed a significant decrease in BOP. *Conclusions:* Three surgical therapies presented favorable results for increasing the KM surrounding implants. Compared with the FGG group, the CM showed similar results in increasing the KM around the dental implants with peri-implantitis.

## 1. Introduction

Following tooth extraction, the remaining keratinized mucosa (KM) is typically narrowed. The muscle attachments, particularly those of the vestibular mentalis, lingual mylohyoideus, and genioglossus, become more prominent; these are, of course, associated with a shallow vestibule [1,2]. Although the importance of KM has been reported, it is currently a controversial issue. Gingival recession is generally reported in the early phase (6–12 months after prosthesis delivery) and may be more pronounced at the sites without KM [3]. The presence of KM is associated with less marginal recession [4]. A systemic review has shown that plaques are easily accumulated at the narrow KM [5]. However, they are not significantly affected by the quality or mobility of the marginal tissue [6]. Preclinical studies have observed inflammatory cell infiltration in both narrow/mobile and wide/firm attached tissues [7,8]. Kennedy et al. reported that the increase in KM did not increase the degree of periodontal health [9].

The requirement of a specific zone of KM surrounding the dental implants to maintain peri-implant health has been a controversial issue. The lack of adequate KM surrounding implants results in a higher plaque accumulation and gingival inflammation and more average annual bone loss [10]. The capability to control plaque accumulation around the implants was better in the presence of KM, emphasizing the critical nature of an adequate KM zone [11]. However, previous studies with multiple regression analysis have not demonstrated any statistically significant association between KM width and peri-implant parameters (plaque, gingival index, or bleeding on probing (BOP)) [12,13].

Several procedures have been used for increasing KM. Apically positioned flap (APF), collagen matrix (CM), and grafting procedures are the major methodologies used for obtaining increased KM. APF is a periodontal surgery in which the entire soft tissue is positioned in the apical direction. The advantages of this procedure are as follows: (1) minimum pocket depth, (2) minimum bone loss due to the prevention of bone exposure, and (3) no more gingival removal during surgery. Grafting procedures use palatal gingiva and other materials to increase the KM of other oral cavity areas. Free gingival graft (FGG) is a predictable method because it maintains the original characteristics as a firm mucosa of the palate [14]. A previous study has reported that APF and vestibuloplasty in combination with FGG were the most effective procedures for increasing the keratinized tissue [15]. However, the disadvantages of this procedure include pain and bleeding during surgery [14]. CM has been considered as an alternative to overcome the disadvantages of FGG.

The relationship between the presence of KM and peri-implant mucositis (peri-implantitis) remains controversial. Peri-implant mucositis and peri-implantitis are inflammatory processes that include gingival swelling, redness, and alveolar bone loss [16]. Although some studies have reported higher rates of peri-implant mucositis at the implants surrounded by or lacking an inadequate KM width (< 2 mm), some other studies have found no association or a positive association between KM width and peri-implant mucositis [17,18]. However, evidence regarding the effect of KM on the long-term health of the peri-implant tissue remains equivocal. Therefore, peri-implant mucositis and peri-implantitis may be initiated by the absence of KM [19,20]. To enhance KM, several techniques, including APF, CM, and FGG, have been clinically used [21,22]. Lin et al. reported that these procedures led to mature and stable soft tissues [4]. However, whether KM has a significant role in treating peri-implantitis remains unknown. Therefore, the purpose of this study was to assess the clinical outcomes of three different surgical therapies (CM, APF, and FGG) for increasing KM surrounding dental implants with peri-implantitis.

## 2. Materials and Methods

### 2.1. Study Population

The study was conducted as a pilot clinical trial. The Research Ethics Committee of Jukjeon Dental Hospital, College of Dentistry, Dankook University (Approval No. 201910-001-001; approved date: 12 November 2018) approved the study’s design and procedures. We performed all protocols according to the Declaration of Helsinki and the Consolidated Standards of Reporting Trials 2010 statements. We obtained written informed consent from all the patients who voluntarily participated in the study. In total, 16 patients who visited the Department of Periodontics of Jukjeon Dental Hospital, College of Dentistry, Dankook University, between November 2018 and February 2020 were involved in this study. We included 20 peri-implantitis sites with an absence of KM (less than 2 mm) in this study. Additionally, we excluded any patients with implant mobility, no panoramic images, as well as those who did not agree to participate in this study. Table 1 shows the general characteristics (e.g., age, gender, and jaw position) of the participants.

### 2.2. Sample Size Calculation

The design of this study is a pilot test. Previous studies were involved in calculating adequate power involving buccal soft tissue augmentation [23,24]. The sample size per group was calculated to detect a standard deviation with 80% power and a 5% alpha error. The required minimum sample size was 7 subjects, and the target sample size was set at 21 subjects for cluster analysis.

### 2.3. Allocation and Surgical Procedures

The experimental sites were divided into three subgroups:

CM group: seven implant sites–porcine CM (Mucograft^®^; Geistlich Pharma AG, Wolhusen, Switzerland) (Figure 1A–F).

APF group: eight implant sites–APF (Figure 1G–L).

FGG group: five implant sites–autologous FGG from the palate (Figure 1M–R).

All treatments were performed by an experienced periodontist (J.T.L). The oral cavity was rinsed with 0.12% chlorhexidine gluconate (Hexamedin^®^, Bukwang Pharmaceutical, Ansan, Korea) for 1 min. After the administration of 2% lidocaine (1:100,000 epinephrine^®^, Huons Co., Ltd., Sungnam, Korea) to induce local anesthesia, we made a partial-thickness flap to expose the periosteum. The detached flap was positioned in the apical direction of the vestibule and fixed by the periosteal suture with resorbable sutures (4-0 Vicryl^®^, Ethicon, Somerville, NJ, USA) to create APF and the recipient bed for FGG and CM. The granulation tissue surrounding the dental implants was removed by the periodontal hand instrument. The implant surface was debrided by using the peri-implantitis treatment kit (IM CURE Kit^®^; Osstem Implant Co., Ltd. Seoul, Korea). For the FGG group, autogenous-free gingiva (thickness of 1.5–2.0 mm) was harvested from the palatal area. FGG was fixed on the recipient bed with cross-mattresses and interrupted sutures. The CM from porcine was also fixed in a similar manner. Amoxycillin and clavulanic acid 375 mg (Augmentin^®^, Ilsung Pharma Co., Seoul, Korea), naproxen sodium (Anaprox^®^, Jongeun Dang Pharmceutical, Co., Seoul, Korea), and almagate (Almagel^®^, Yuhan Pharma Co., Seoul, Korea) were prescribed to all patients three times daily for 1 week. All the patients were instructed to rinse their mouth twice daily with 0.12% chlorhexidine gluconate (Hexamedin^®^, Bukwang Pharmaceutical, Ansan, Korea) for 30 seconds for 1 week. Sutures were removed 2 weeks after surgery.

### 2.4. Clinical Assessment

An experienced periodontist (J.T.L) measured all clinical parameters (probing pocket depth (PPD) and BOP) at six sites per implant using a periodontal probe (PCP-12, Hu-Friedy, Rotterdam, Netherlands). The initial examination was performed before surgery (T0). Follow-up examinations were conducted at the following time points: 30 (T1), 60 (T2), 90 (T3), and 180 (T4) days after the surgery (Figure 2). The horizontal/vertical KM width (vertical KM width: a distance from the free gingival margin to MGJ (mucogingival junction); horizontal KM width: a distance between the extension lines in the apical direction at the mesial and distal margin of the implant prosthesis) was measured at the buccal aspect of each implant, whereas the augmented sites were evaluated in terms of their clinical appearances (texture and color) using the clinical photo by two independent investigators (H.S.C and J.T.L). The method was a modification of the method reported by Lim et al. [15]. Regarding peri-implantitis disease classification, we used a cone-beam computed tomography (CBCT) scanner (Kodak 9500^®^, Carestream Health, Rochester, NY, USA) in this study, which provides a grayscale image of 14 bits with a voxel size of 0.2 mm per side. The CBCT images were viewed using 3D imaging software (OnDemand 3D, Cybermed Co., Seoul, Korea). Panoramic and intraoral radiographic X-ray was also used. Based on these data, all implants were classified according to previous studies [25,26]. Two investigators (H.S.C and J.T.L) also evaluated the classification of each implant.

### 2.5. Statistical Analysis

Each site was considered as a statistical unit for statistical analysis. The mean width of the keratinized tissue was calculated at each measuring point: T0, T1, T2, T3, and T4, and the collected data (mm) were presented as mean ± standard deviation. Statistical analyses were performed using the SPSS Statistics 23.0 software (IBM Corp., Armonk, NY, USA). The Wilcoxon signed-rank test was used for intragroup comparisons (T0 vs. T1 and T2 vs. T3), whereas the Kruskal–Wallis test with post-hoc Bonferroni correction was used for intergroup comparisons. Statistical significance was set at *p* < 0.05 for this study.

## 3. Results

### 3.1. Baseline Characteristics

Twenty implant sites with peri-implantitis from 16 patients were divided into the CM, APF, and FGG groups. Table 1 shows the general characteristics of the patients. The mean ages of each group were as follows: CM: 60.43 (4.61), APF: 63.88 (4.03), and FGG: 58.00 (3.83). The implant numbers of the upper jaw were one (CM), six (APF), and two (FGG), whereas those of the lower jaw were six (CM), two (APF), and three (FGG). Depending on the bony defect, there were four types of peri-implantitis in the CM group (Class Id: 1, Class Ib + II: 1, Class Ic + II: 2, and Class Id + II: 3). Five types of peri-implantitis were observed in the APF group (Class Ib: 1, Class Id: 1, Class Ic + II: 2, Class Id + II: 2, and Class Ie + II: 2). The FGG group had three types of peri-implantitis (Class Ib: 2, Class Ic + II: 1, and Class Id + II: 2). The systemic diseases were osteoporosis, heart disease, and hypertension, and none of these diseases were found in any of the groups.

### 3.2. Width of KM

KM width comprised two parts: horizontal and vertical widths (Table 2 and Figure 3).

#### 3.2.1. Horizontal Width

The horizontal width increased from T0 and was maintained up to T4 in the CM group (T0 = 1.14, T1–T4 = 4.71). There were significant differences in each period in the CM group (*p* = 0.004 (T0 vs. T1), 0.002 (T0 vs. T2), 0.004 (T0 vs. T3), 0.004 (T0 vs. T4), for each). The APF group showed an increase in horizontal KM; however, the increase was not significantly different between T0 and other periods (T1–T4) (*p* = 0.447, 0.327, 0.871, 0.914, respectively). In the FGG group, compared with T0, an increased KM value was observed in each period close to significant differences (*p* = 0.055 (T0 vs. T1), 0.072 (T0 vs. T2), 0.043 (T0 vs. T3), 0.068 (T0 vs. T4), for each).

#### 3.2.2. Vertical Width

All the three groups had significant increases in vertical KM between T0 and T1 (*p* = 0.003 (CM), 0.002 (APF), 0.008 (FGG)). In the CM and FGG groups, the vertical KM length increased until T2 and then gradually decreased (CM: 0.43 (T0), 3.71 (T1), 4.14 (T2), 3.86 (T3), and 3.86 (T4) and FGG: 0.80 (T0), 4.40 (T1), 5.00 (T2), and 4.50 (T4)]). APF also showed an increase in vertical KM from T0 to T3 (0.88 (T0), 2.71 (T1), 2.57 (T2), 3.00 (T3)). There was a decrease in KM height between T3 and T4 in APF (3.00 (T3), and 2.50 (T4)).

### 3.3. Probing Pocket Depth (PPD)

The analysis was performed based on the number of probing sites (six whole sites and three buccal sites) because the CM, APF, and FGG procedures were mainly performed on the buccal gingiva (Table 2 and Figure 3).

#### 3.3.1. Six Whole Sites

The differences in PPD values between T0 and T1 were 0.054 (CM), 0.032 (APF), and 0.009 (FGG). The *p*-value between T1 and T2 in the APF and FGG groups had significant differences except for that in the CM group (*p* = 0.222 (CM), *p* = 0.048 (APF), and *p* = 0.046 (FGG)). At T3 and T4, only the FGG group had significant differences in PPD values as compared with those at T0.

#### 3.3.2. Three Buccal Sites

The *p*-values of the three groups between T0 and T1 were 0.082 (CM), 0.053 (APF), and 0.008 (FGG). There were no statistically significant differences observed between T0 and T2 (*p* = 0.190 (CM), *p* = 0.114 (APF), and *p* = 0.065 (FGG)). However, in the FGG group, a significantly different value was observed at T3 and T4 than at T0 (T3: *p* = 0.022 and T4: *p* = 0.044). There was no difference in the other groups at T2, T3, and T4.

### 3.4. Bleeding on Probing (BOP)

Table 2 and Figure 3 also provide a summary of BOP. The analysis also comprised subgroups of six whole sites and three only buccal sites.

#### 3.4.1. Six Whole Sites

Two groups (APF and FGG) showed significant differences at T1 than at T0 (*p* =0.006 (APF) and *p* =0.050 (FGG)). Although bleeding was reduced in the CM group, there was no significant difference in parameters between T0 and the other time points (T1–T4; T0 = 3.29, T1 = 1.43, T2 = 1.14, T3 = 1.57, and T4 = 2.43).

#### 3.4.2. Three Buccal Sites

There was no difference in BOP in the CM group at the T1–T4 periods. The amount of bleeding in the CM group was lower at T1–T4 than at T0; however, there was no significant difference in the CM group compared with the APF and FGG group.

### 3.5. Bone-Level Change around Implant

Alveolar bone-level change was assessed using radiographic methods (panorama, intraoral radiographic X-ray, and CBCT). In the three groups, severe alveolar bone resorption was not observed between T0 and T4 (Figure 4).

## 4. Discussion

This is a comparative study of the clinical results of three different surgical therapies (CM, APF, and FGG) for increasing KM surrounding dental implants with peri-implantitis. Patients who first visited the dental hospital complained of pain or bleeding around the implant. After surgical treatment, a decrease in pain or bleeding was observed in the patients in this study. All groups had significant differences with respect to increasing horizontal and vertical KM. The CM and FGG groups had more KM than the APF group. There was no significant difference in parameters between the two groups (CM vs. APF groups, CM vs. FGG groups, and APF vs. FGG groups) at T0–T3. The measured PPD significantly differed between the APF and FGG groups at T1 and T2. Bleeding in the three groups was reduced at T1–T4 compared with that at T0. APF and FGG showed significant differences in BOP. In this study, PPD showed a decrease in the three groups. The APF and FGG groups showed significant differences in PPD at T1 and T2. At T3 and T4, only FGG showed significance. This result is consistent with that of a previous study that showed a decrease in PPD after the grafting of soft tissues surrounding implants with peri-implantitis [27]. There was no significant decrease in PPD in the CM group. Overall, BOP also tended to decrease in all groups in this study, indicating that inflammation was ameliorated after the treatment of peri-implantitis. However, no significant differences were observed in the CM group at all the periods other than at T0.

One reason for these results is that severe bone loss of the buccal area contributes to unstable fixation of the CM. Ravida et al. demonstrated that the results in surgical peri-implantitis treatment were not affected by the presence of KM but by the severity of bone loss at the time of treatment [27]. Other previous studies that used similar materials considered the surgical therapeutic outcomes as good because most buccal bones were intact [15,28]. However, dental implants in this study had lost their buccal bones. Additionally, vertical bone loss was also observed (peri-implantitis classification types in this study: Class Ib, Class Id, Class Ib + II, Class Ic + II, Class Id + II, and Class Ie + II). Therefore, it is hard to attach the CM to the narrow buccal region of the dental implant. Because CM has a relatively thin thickness and weak mechanical property than autogenous gingival graft, it might not be easy to attach CM, as compared with FGG. In this study, suppuration occurred in the CM group after treatment in one case. Therefore, attempts to increase KM around implants with advanced bone loss might lead to less favorable results than those with the intact bone.

Another factor affecting the outcome of this study was gingival rebound by muscle pull. Most implant sites involved in this study were positioned in the mandibular posterior (molar) area; therefore, it is considered that a rebound tendency frequently occurred at this area compared to other areas [29]. A previous study reported that several factors (such as low vestibule and high muscle pull) diminish the effect of CM [15]. The results of this study explain that the unstable fixation of CM is related to gingival rebound, low vestibule, and high muscle pull. Therefore, two things are recommended when CM is used on the peri-implant area: (1) adequately remained buccal bone is needed to fix CM, and (2) pre-surgical procedure is required to increase the vestibule area before CM application.

A previous study demonstrated that APF/vestibuloplasty plus FGG/SCTG was the most successful method to increase KM width (1.4–3.3 mm). However, APF/vestibuloplasty plus CM reported less gain in KM. Additionally, the soft tissue graft improved the gingival thickness and esthetics, as compared with the non-grafted soft tissue [30]. This study also showed a horizontal and vertical increase in KM (horizontal and vertical: FGG > CM > APF), which is similar to the result of a previous study [15]. In other studies, autogenous tissue graft has shown superior properties over CM [31,32]. In a histologic analysis in a previous study, the CM group had more inflammatory infiltration and less vascularization and fibroblasts than the FGG group after 10 days of the graft procedure. Delayed healing in the CM group was also observed during the 60-day follow-up period [31]. Shrinkage tendency was shorter for the CM material, and CM showed various ranges of shrinkage ratios than FGG (de Resende et al. 2019: CM: 56% vs. FGG: 12%; Wei et al. 2000: CM 71% vs. FGG: 16%) [31,32]. However, CM has the advantage of superior esthetic perception and easy grafting procedure by non-preparation of the donor site. Autologous soft tissue grafting resulted in postoperative pain, bleeding, and swelling at the palatal donor site. Burkhardt et al. reported that the visual analog scale value of pain was 4.1 at the palatal donor site 1 day after soft tissue graft surgery [33]. The CM procedure does not require additional surgery that will induce pain to obtain autologous soft tissue. Reduced surgery time is another strength of using the CM procedure.

Increasing the KM of the peri-implant is good for the long-term stability of the dental implant. Schmitt et al. have demonstrated that FGG and CM are suitable for regenerating KM surrounding the peri-implant tissue with sufficient long-term stability [28]. In a previous study, 11 of 39 dental implants (28.2%) were removed in KM width less than 2 mm, whereas only 1 out of 29 dental implants was lost. A shorter follow-up period was shown in the implant site with a KM width less than 2 mm [27]. However, unstable fixation of soft tissue graft (CM, APF, and FGG) by bone loss with peri-implantitis appears to reduce the long-term stability. Therefore, early detection of peri-implantitis and procedures to increase KM are recommended for long-term stability.

The sample size in this study was insufficient to evaluate all the groups. If the cohort study was larger, it would be considered that the results may be better than those obtained in this study. It would be necessary to match the surgical sites and systemic diseases among groups in future studies. Furthermore, 6 months were insufficient to observe KM maintenance; therefore, a long-term follow-up period is required. There are a limited number of studies that have evaluated the relationship between peri-implantitis and KM. Despite these limitations, this study is relevant because it explores the relationship between advanced peri-implantitis and an increase in KM.

## 5. Conclusions

Three surgical therapies (CM, APF, and FGG) demonstrated favorable results for increased KM around the dental implants with peri-implantitis. As compared with FGG, CM showed similar results in increasing the KM.

## Figures and Tables

**Figure 1 medicina-57-01093-f001:**
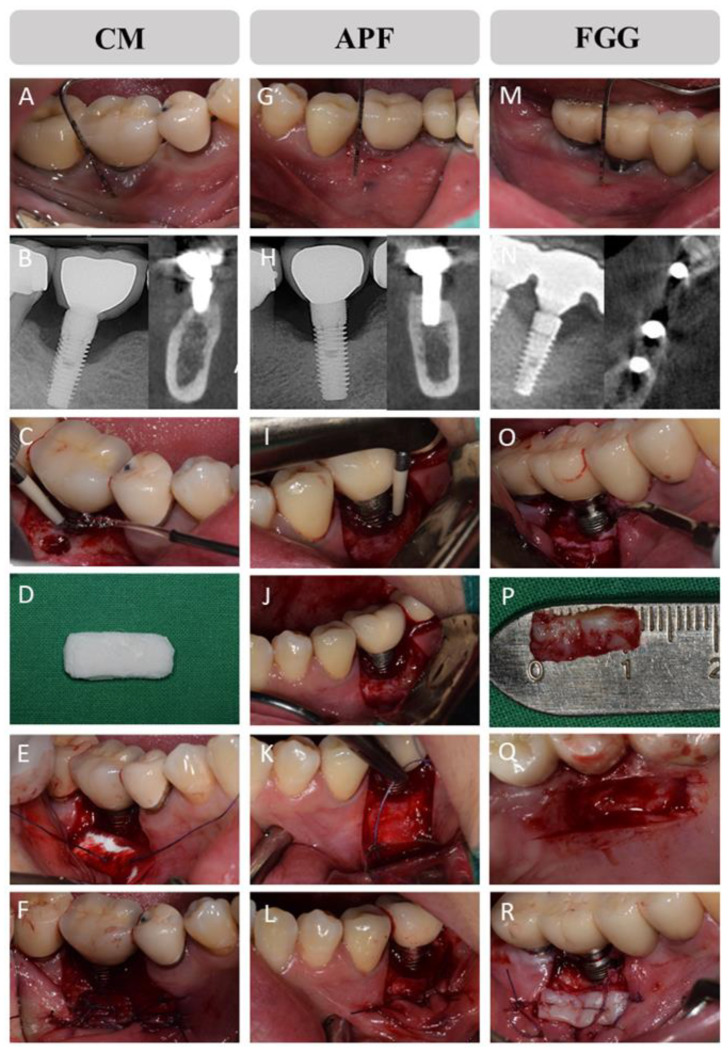
Clinical photographs for three different surgical treatments for increasing keratinized mucosa (KM) around the dental implant with peri-implantitis. CM (**A**–**F**): porcine collagen matrix, APF (**G**–**L**): apically positioned flap, FGG (**M**–**R**): autologous free gingival graft from the palate.

**Figure 2 medicina-57-01093-f002:**
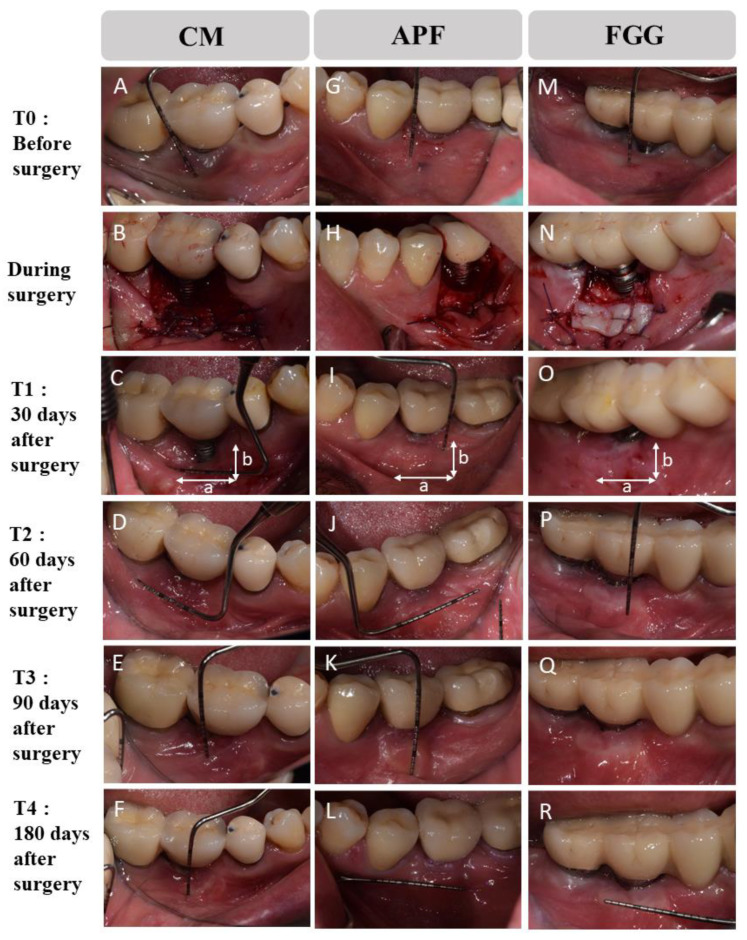
Clinical photographs at the following time points: before surgery (T0), 30 (T1), 60 (T2), 90 (T3), and 180 (T4) days after surgery. CM (**A**–**F**): porcine collagen matrix, APF (**G**–**L**): apically positioned flap, FGG (**M**–**R**): autologous free gingival graft from the palate. a: horizontal width of keratinized mucosa (KM): a distance between the extension lines in the apical direction at the mesial and distal margin of the implant prosthesis. b: vertical width of KM: a distance from the free gingival margin to MGJ (mucogingival junction)

**Figure 3 medicina-57-01093-f003:**
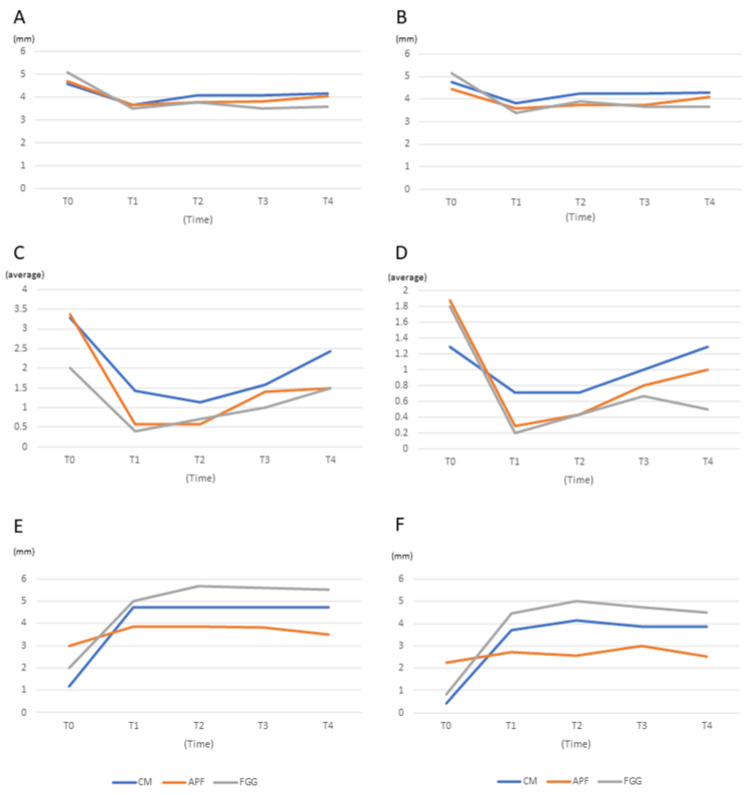
Change of clinical parameters (probing pocket depth (PPD), bleeding on probing (BOP)) and width of keratinized mucosa (KM) at the following time points: before surgery (T0) and 30 (T1), 60 (T2), 90 (T3), and 180 (T4) days after surgery. CM: porcine collagen matrix, APF: apically positioned flap, FGG: autologous free gingival graft from the palate. (**A**): probing pocket depth (PPD) on the whole six sites around the implant, (**B**): probing pocket depth (PPD) on the three buccal sites around the implant, (**C**): bleeding on probing (BOP) on the whole six sites around the implant, (**D**): bleeding on probing (BOP) on the three buccal sites around the implant, (**E**): horizontal width of keratinized mucosa (KM), (**F**): vertical width of KM.

**Figure 4 medicina-57-01093-f004:**
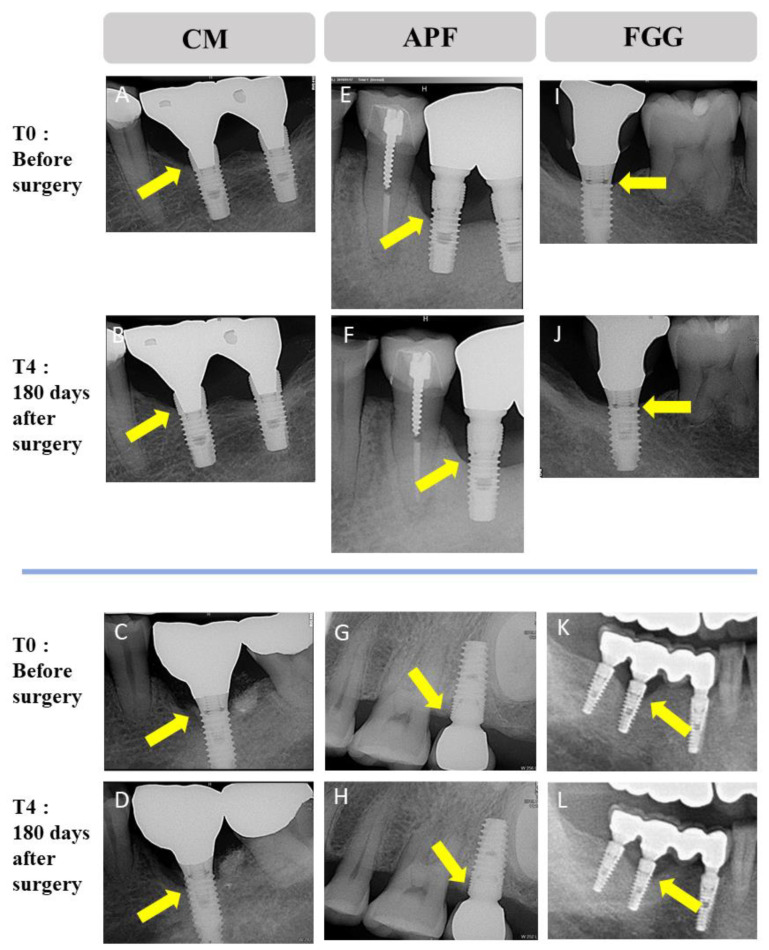
Change of alveolar bone at the following time points: before surgery (T0) and 180 (T4) days after surgery. There was no severe alveolar bone-level change around implants in three groups (CM, APF and FGG). CM: porcine collagen matrix, APF: apically positioned flap, FGG: autologous free gingival graft from the palate.

**Table 1 medicina-57-01093-t001:** General characteristics.

Characteristics	CM	APF	FGG
**N**	7	8	5
**Age (years) (Mean (SD)**	60.43 (4.61)	63.88 (4.03)	58.00 (3.83)
**Sex**			
Male	4	4	3
Female	3	4	2
**Jaw position**			
Upper	1	6	2
Lower	6	2	3
**Oral regions**			
Anterior teeth	1	1	
Premolars		1	
Molars	6	6	5
**Classification of peri-implantitis**			
Class Ib		1	2
Class Id	1	1	
Class Ib + II	1		
Class Ic + II	2	2	1
Class Id + II	3	2	2
Class Ie + II		2	
**Systemic disease**			
Osteoporosis	2	0	1
Heart disease	2	1	0
Hypertension	2	3	0
None	1	4	4
**Medicine**			
Osteoporosis	2	2	1
Anticoagulant	2	0	0
Hypertension	2	1	0
None	1	5	4
**Smoking**			
Now	0	2	0
Past	1	1	1
Not	6	5	4

Values are presented as mean (standard deviation). CM: porcine collagen matrix, APF: apically positioned flap, FGG: autologous free gingival graft from the palate.

**Table 2 medicina-57-01093-t002:** Clinical parameters.

	CM	APF	FGG		CM	APF	FGG		CM	APF	FGG
KM				PPD				BOP			
Horizontal width				6 sites				6 sites			
T0	1.14 (1.95)	3.00 (2.07)	2.00 (2.35)	T0	4.57 (0.86)	4.71 (0.81)	5.07 (0.35)		3.29 (2.56)	3.38 (2.13)	2.00 (1.23)
T1	4.71 (0.95) ^‖^	3.86 (0.90)	5.00 (1.00)	T1	3.64 (0.58)	3.64 (0.78) ^‖^	3.50 (0.59) ^‖^		1.43 (1.13)	0.57 (0.79) ^‖^	0.40 (0.55) ^‖^
T2	4.71 (0.49) ^‖^	3.86 (1.22)	5.67 (0.58)	T2	4.07 (0.71)	3.79 (0.67) ^‖^	3.78 (0.95) ^‖^		1.14 (1.46)	0.57 (1.13) ^‖^	N/A ^‖^
T3	4.71 (0.76) ^†,‖^	3.80 (1.10) *^,‡^	N/A ^†,‖,§^	T3	4.10 (0.72) ^¶^	3.80 (0.77)	3.50 (0.67) ^‖^		1.57 (1.51)	1.40 (0.90) ^‖^	1.00 (1.00)
T4	4.71 (0.76) ^†,‖^	3.50 (1.23) *^,‡^	5.50 (0.71) ^†,§^	T4	4.17 (0.75) ^¶^	4.06 (0.82)	3.58 (1.06) ^‖^		2.43 (1.99)	1.50 (1.05) ^‖^	1.50 (0.71)
Vertical width				Buccal 3 sites				Buccal 3 sites			
T0	0.43 (0.79)	0.88 (0.64)	0.80 (0.84)	T0	4.76 (1.05)	4.46 (0.87)	5.13 (0.30)		1.29 (1.60)	1.85 (0.99)	1.80 (1.30)
T1	3.71 (1.60) ^‖^	2.71 (0.95) ^‡,‖^	4.40 (0.55) ^§^	T1	3.81 (0.74)	3.57 (0.79)	3.40 (0.83) ^‖^		0.71 (0.76)	0.29 (0.49) ^‖^	0.2 (0.45) ^‖^
T2	4.14 (1.35) ^†,‖^	2.57 (1.13) *^,‡,‖^	5.00 (1.00) ^†,‖,§^	T2	4.24 (0.88)	3.76 (0.66)	3.90 (1.02)		0.71 (0.95)	0.43 (0.79) ^‖^	N/A ^‖^
T3	3.86 (1.22) ^†,‖^	3.00 (0.71) *^,‡,‖^	N/A ^‖,§^	T3	4.24 (0.88)	3.73 (0.68)	3.67 (0.88) ^‖^		1.00 (0.82)	0.80 (0.45) ^‖^	0.67 (0.58)
T4	3.86 (1.22) ^†,‖^	2.50 (0.55) *^,‡,‖^	4.50 (0.71) ^†,‖,§^	T4	4.29 (0.89)	4.11 (0.81)	3.67 (0.94) ^‖^		1.29 (1.11)	1.00 (0.63) ^‖^	0.50 (0.71) ^‖^

Values are presented as mean (standard deviation). CM: porcine collagen matrix, APF: apically positioned flap, FGG: autologous free gingival graft from the palate KM: keratinized mucosa, PPD: probing pocket depth, BOP: bleeding on probing. T0: before treatment, T1: 30 days after treatment, T2: 60 days after treatment, T3: 90 days after treatment, and T4: 180 days after treatment. * Statistical significance (*p* < 0.05); the Kruskal–Wallis test with post-hoc Bonferroni correction was used for intergroup comparisons vs. CM. † Statistical significance (*p* < 0.05); the Kruskal–Wallis test with post-hoc Bonferroni correction was used for intergroup comparisons vs. APF. ‡ Statistical significance (*p* < 0.05); the Kruskal–Wallis test with post-hoc Bonferroni correction was used for intergroup comparisons vs. FGG. § Statistical significance (*p* < 0.05); the Kruskal–Wallis test with post-hoc Bonferroni correction was used for intergroup comparisons in CM, APF, and FGG groups. ‖ Statistical significance (*p* < 0.05); the Wilcoxon signed-rank test was used for intragroup comparisons vs. T0. ¶ Statistical significance (*p* < 0.05); the Wilcoxon signed-rank test was used for intragroup comparisons vs. T1. # Statistical significance (*p* < 0.05); the Wilcoxon signed-rank test was used for intragroup comparisons vs. T2. ** Statistical significance (*p* < 0.05); the Wilcoxon signed-rank test was used for intragroup comparisons vs. T3. # Statistical significance (p < 0.05); the Wilcoxon signed-rank test was used for intragroup compar-isons vs. T2. ** Statistical significance (p < 0.05); the Wilcoxon signed-rank test was used for intragroup comparisons vs. T3.

## Data Availability

All available data are presented within the article or are available on request from the corresponding author.

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
