# Peer review of "The Effect of Three Surgical Therapies to Increase Keratinized Mucosa Surrounding Dental Implants with Peri-Implantitis: A Pilot Study"

_medicina, 2021, doi:10.3390/medicina57101093_

Round 1

Reviewer 1 Report

authors Could include more patients for stronger statistical relevance.  It the clinical results are important for the management of peri implantitis caused impart by lack of KG

Author Response

Comments: authors Could include more patients for stronger statistical relevance.  It the clinical results are important for the management of peri -implantitis caused impart by lack of KG

Answer:

Thanks for your kind comment, I added sentences below in discussion

'Patients who first visited the dental hospital complained of pain and bleeding around the implant. After surgical treatment, a decrease in pain and bleeding was observed in patients.'

'The sample size in this study was insufficient to evaluate all the groups. If the cohort study is larger, it is considered that the results may be better than those obtained in this study. It would be necessary to match the surgical sites and systemic diseases among groups in future study. Furthermore, 6 months were insufficient to observe KM maintenance; therefore, a long-term follow-up period is required.'

Reviewer 2 Report

Main Comments

As a treatment method for peri-implantitis, it is an article showing the effectiveness of CM by comparing CM with APF and FGG over time, and it is considered to have high clinical significance as a pilot study.

However, it is necessary to consider that there are characteristics with large differences among the three groups regarding patient selection, and that there is no information on the most important bone loss (gain) despite evaluation over time.

Minor Comments

  1. 1. Regarding patient selection, CM has 6 implants out of 7 implants in the mandible, while APF has 6 implants out of 8 implants in the maxilla. There are also differences between groups regarding bone defects in Peri-implantitis, differences in sites, and differences between groups regarding systemic disease and smoking. Since these are considered to have a great influence on the results, please unify the site and defect morphology or describe the limitations and problems of this study in Discussion.
  2. There is no data on bone loss (gain) over time. Please add fig as it is important data for evaluating the treatment result of peri-implantitis.

Author Response

1. Regarding patient selection, CM has 6 implants out of 7 implants in the mandible, while APF has 6 implants out of 8 implants in the maxilla. There are also differences between groups regarding bone defects in Peri-implantitis, differences in sites, and differences between groups regarding systemic disease and smoking. Since these are considered to have a great influence on the results, please unify the site and defect morphology or describe the limitations and problems of this study in Discussion.

Answer: 

Thanks for your kind comment, I add sentences below,

'The sample size in this study was insufficient to evaluate all the groups. If the cohort study is larger, it is considered that the results may be better than those obtained in this study. It would be necessary to match the surgical sites and systemic diseases among groups in future study. Furthermore, 6 months were insufficient to observe KM maintenance; therefore, a long-term follow-up period is required.'

2. There is no data on bone loss (gain) over time. Please add fig as it is important data for evaluating the treatment result of peri-implantitis.

Thanks for your kind comment. Since this study focused on changes in KM, data on bone loss and increase were not separately evaluated. However, since each patient has radiographic x-ray data, we have included a figure 4 and sentence [‘Alveolar bone loss was assessed using radiographic methods (panorama, intraoral radiographic X-ray and CBCT). In the three groups, severe alveolar bone resorption was not observed between T0 and T4 (Figure 4).’] for this. There were not many changes in alveolar bone in all groups.

Reviewer 3 Report

The manuscript is very well idealized. Its scientific contribution is going to be very interesting for periodontologists and oral surgeons. I recommand the acceptance in the way it is. 

Author Response

The manuscript is very well idealized. Its scientific contribution is going to be very interesting for periodontologists and oral surgeons. I recommand the acceptance in the way it is. 

Answer: 

Thanks for your kind comment, I will re-submit this article through additional reinforcement.

Round 2

Reviewer 2 Report

The problem is that the number of samples is small and the research design is not unified, but since it is an interesting content for a pilot study, it is accepted.